# Single-cell RNA-seq of rheumatoid arthritis synovial tissue using low-cost microfluidic instrumentation

William Stephenson [1], Laura T. Donlin[2,3], Andrew Butler [4,5], Cristina Rozo[2], Bernadette Bracken[4,5], Ali Rashidfarrokhi[4,6], Susan M. Goodman[2,3], Lionel B. Ivashkiv [2,3], Vivian P. Bykerk[2,3], Dana E. Orange[2,7], Robert B. Darnell[4,7,8], Harold P. Swerdlow [1] & Rahul Satija [4,5]

Droplet-based single-cell RNA-seq has emerged as a powerful technique for massively parallel cellular profiling. While this approach offers the exciting promise to deconvolute cellular heterogeneity in diseased tissues, the lack of cost-effective and user-friendly instrumentation has hindered widespread adoption of droplet microfluidic techniques. To address this, we developed a 3D-printed, low-cost droplet microfluidic control instrument and deploy it in a clinical environment to perform single-cell transcriptome profiling of dis-aggregated synovial tissue from five rheumatoid arthritis patients. We sequence 20,387 single cells revealing 13 transcriptomically distinct clusters. These encompass an unsupervised draft atlas of the autoimmune infiltrate that contribute to disease biology. Additionally, we identify previously uncharacterized fibroblast subpopulations and discern their spatial location within the synovium. We envision that this instrument will have broad utility in both research and clinical settings, enabling low-cost and routine application of microfluidic techniques.

[1] Technology Innovation Lab, New York Genome Center, New York, NY 10013, USA. [2] Hospital for Special Surgery, New York, NY 10021, USA. [3] Weill Cornell Medical College, New York, NY 10065, USA. [4] New York Genome Center, New York, NY 10013, USA. [5] New York University, Center for Genomics and Systems Biology, New York, NY 10003, USA. [6] New York University School of Medicine, New York, NY 10016, USA. [7] Laboratory of Neuro-Oncology, Rockefeller University, New York, NY 10065, USA. [8] Howard Hughes Medical Institute, Rockefeller University, New York, NY 10065, USA. William Stephenson, Laura T. Donlin and Andrew Butler contributed equally to this work. Harold P. Swerdlow and Rahul Satija jointly supervised this work. Correspondence and requests for materials should be addressed to W.S. (email: wstephenson@nygenome.org) or to R.S. (email: rsatija@nygenome.org)

The complex architecture and associated higher-order function of human tissues relies on functionally and molecularly diverse cell populations. Disease states represent significant perturbations to cellular heterogeneity, with tissue-resident cells acquiring altered phenotypes and circulating cells infiltrating into the tissue. Therefore, defining the cellular subsets found in pathologic tissues provides insights into disease etiology and treatment options. Traditional methods such as flow cytometry, which require a priori knowledge of cell type-specific markers, have begun to define this landscape, but fall short in comprehensively identifying cellular states in a tissue, with particular difficulty detecting extremely rare subpopulations.

Technological advancements in automation, microfluidics, and molecular barcoding schemes have permitted the sequencing of single cells with unprecedented throughput and resolution[1–4]. In particular, recent studies featuring analysis of $10^4$–$10^5$ single cells have enabled unbiased profiling of cellular heterogeneity, where entire tissues can be profiled without advance enrichment of individual cell types[1,5,6]. In spite of this progress, technological advances can be slow to permeate into resource-limited clinical arenas due to a variety of reasons related to cost, personnel requirements, space or infrastructure. Specifically, a major barrier to widespread adoption of droplet microfluidic techniques is the lack of cost-effective and reliable instrumentation[7,8]. Microfluidic experiments are typically performed using commercial instruments which are expensive and often configured for a single purpose, or custom research instrument setups which are comprised of multiple pieces of equipment and rarely portable. Particularly in clinical settings, microfluidic instrumentation is not always proximal to the site of cell sample generation requiring transport to external sites or cell preservation, both of which can alter cellular transcriptomes or result in extensive cell death[6,9].

To address these short-comings and provide a low-cost option for single-cell transcriptome profiling, we have developed an open-source portable instrument for performing single-cell droplet microfluidic experiments in research and clinical settings. Recent microwell-based transcriptome profiling approaches have been shown to be advantageous for low-cost portable transcriptome profiling[10–12], however some of these techniques are challenging to perform and or require extensive chemical modification to fabricate the devices. Additionally, the fixed architecture of microwell (partitioning) microfluidic devices dictates their use for specific applications. In contrast, the platform presented here is easy to use and can be implemented for a variety of droplet microfluidic (partitioning) or continuous phase microfluidic based experiments. Potential applications of this system include recent work profiling immune repertoires from hundreds of thousands of single cells[13] and combined single-cell transcriptome and epitope profiling[14] in addition to ddPCR[15], ddMDA[16], hydrogel microsphere fabrication for 3D cell culture[17,18], chemical microfluidic gradient generation[19] and microparticle size sorting[20–22]. The instrument is comprised of electronic and pneumatic components affixed to a 3D printed frame. The entire system is operated through software control using a graphical user interface on a touchscreen. Requiring only a standard wall power outlet, the instrument has an extremely small footprint; small enough to fit on a bench top or in a bio-containment hood. The total cost of materials to construct an instrument is approximately $575. This represents an approximately 20-fold and 200-fold reduction in cost compared to a research-level, syringe-pump based microfluidic setup, and a commercial microfluidic platform, respectively.

We applied the microfluidic control instrument in conjunction with the Drop-seq technique[1] to perform unbiased identification of transcriptomic states in diseased synovial tissue, which becomes highly inflamed in rheumatoid arthritis (RA) and drives joint dysfunction. RA is a common autoimmune disease affecting approximately 1% of the population. While the cause of RA is not precisely known, disease etiology is hypothesized to originate from a combination of environmental and genetic factors[23,24]. RA affects the lining of the joint; the synovial membrane, leading to painful inflammation, hyperplasia, and joint destruction. RA is clinically characterized by multiple tender and swollen joints, autoantibody production (rheumatoid factor and anti-citrullinated protein antibody or ACPA) in addition to cartilage and bone erosion[25]. Unlike other tissue membranes with an epithelial layer, the synovial lining is composed of contiguously aligned fibroblasts and macrophages 2–3 cells deep[26]. In RA, the membrane lining is expanded to 10 − 20 cells deep and synovial fibroblasts assume an aggressive phenotype marked by the expression of disease relevant cytokines, chemokines and extracellular matrix remodeling factors[27–29]. The sublining is marked by an accumulation of lymphocytes, macrophages, and dendritic cells amidst the subintimal synovial fibroblasts. Pioneering studies have uncovered heterogeneity in fibroblast morphology[30] and phenotype[31,32], observing differences in activation state and invasive behavior[33,34]. In addition, in situ hybridization experiments have identified non-uniform activation of inflammatory drivers and matrix metalloproteinases[35,36], motivating the use of our unsupervised approach to catalog fibroblast subpopulations, and molecular markers which define them.

Here we describe the design of a microfluidic control instrument that can be assembled with 3D-printed and commercial components at low cost, is fully portable, and functions as a reliable and flexible droplet generator. The instrument is fully open-source and instructions for its use and construction have been deposited in the open fluidics repository Metafluidics[37] (https://metafluidics.org/devices/minidrops/). We adapt this device to perform massively parallel single-cell RNA-seq (Drop-seq), observing metrics and performance that are indistinguishable from a research level Drop-seq setup. We deploy this instrument to a hospital laboratory to profile 20,387 single cells from the synovial tissue of 5 RA patients. To our knowledge, this represents the first 'atlas' of hematopoietic and fibroblast transcriptional subtypes from scRNA-seq of autoimmune disease tissue. We identify 13 subpopulations, including both abundant and rare groups that contribute to disease biology, alongside additional sources of heterogeneity within immune clusters. We also define cellular subsets of synovial fibroblasts, validate subset markers using immunofluorescence and flow cytometry, and characterize their spatial distribution in intact tissue. The deconvolution of cellular complexity in a diseased tissue by this portable device provides a template for the application of droplet-based single-cell transcriptome profiling for routine clinical analysis.

## Results

**Development of droplet microfluidic control instrument.** To perform single-cell transcriptome profiling experiments in clinical settings at low cost, the components of a standard Drop-seq setup were replaced with alternative miniature components and packaged onto a multi-tiered 3D printed frame. (Fig. 1a–c, Supplementary Figure 1) For example, syringe pumps in a standard Drop-seq setup (which provide a means for fluid flow through a microfluidic chip) were replaced with components such as a micro air-pump, regulators, and micro solenoid valves. These components are similarly effective for providing adequate fluid flow, in a smaller footprint and at significantly lower cost. Stirring of barcoded microparticles is achieved through actuation of a stepper motor affixed with a permanent magnet at the end of 3D printed shaft. Rotation of the motor shaft locally inverts a

magnetic field thereby tumbling a magnetic stir disc in the microparticle fluid reservoir. A custom printed circuit board (PCB) was designed to interface the electronic and pneumatic components of the instrument to a single board computer (Raspberry Pi). Further critical components of the instrument include pressure sensors for optimal flow rate determination, micro solenoid valves for on-demand pressure actuation, and a microscope for real-time experiment monitoring. The microscope is comprised of an inexpensive 5-megapixel CMOS camera coupled with a laser diode collimating lens. This provides sufficient magnification operating in fixed-focus mode to view the microfluidic channels with the ability to resolve single cells. (Fig. 1d, Supplementary Movie 1) The instrument is operated through a custom graphical user interface on a touchscreen. All components were affixed to a 3D printed frame measuring approximately 21 cm by 20 cm and 9 cm tall (Fig. 1b). For Drop-seq experiments, fluorinated oil, cells, and barcoded microparticles are pipetted into fluid reservoirs situated at the rear of the instrument. Custom pressure caps seal the vial and tubing connections are made to a microfluidic chip situated on the top of the instrument above the microscope camera. The small footprint of the device permitted use in clinical laboratory space requiring only a standard wall outlet for power. In order to facilitate adoption of the device, we have completely "open-sourced" the instrument. Users can find a complete bill of materials (Supplementary Table 1), design files, and documentation required to build and operate the microfluidic control instrument in the supplementary materials and in the Metafluidics repository[37].

**Instrument validation and operation.** To validate the design and operation of the instrument we first assessed the droplets produced by the device in conjunction with a modified Drop-seq microfluidic chip compared to droplets produced by a syringe pump based Drop-seq setup using the original microfluidic chip

design (Supplementary Figure 2). Droplets produced using the instrument displayed a high degree of uniformity (diameter = 105 ± 3 μm) across multiple microfluidic chips and instruments at identical operating pressures indicating stable and reproducible flow patterns of the assembly (Fig. 1e). Next, the loading of microparticles (tested at an optimized concentration) into droplets was assessed using a image analysis software to measure the number of empty, singly occupied, and doubly occupied droplets. The resulting droplet occupancy profile followed a Poisson distribution, as expected for the stochastic loading process described here (Fig. 1f). The instrument processes 1 mL of cells at a concentration of 150–200 cells/μl in about 30 min, generating over one million droplets at a generation rate of approximately 700 Hz.

To compare the technical characteristics of the instrument against a standard syringe pump based Drop-seq setup, we performed benchmark experiments to measure single-cell loading, sensitivity and accuracy metrics. First, to validate single-cell encapsulation, we performed species mixing experiments at two different initial cell input concentrations in which approximately equal numbers of HEK293 (human) cells and NIH 3T3 (mouse) cells were combined in a single run, followed by shallow sequencing to identify mixed-species doublets (Fig. 1g). As observed with a standard Drop-seq setup[1], we found that the degree of observable species mixed droplets is dependent on total cell input concentration, enabling users to identify a input loading concentration to match their desired doublet rate. Next, to explore the mRNA capture sensitivity and accuracy we used synthetic ERCC RNA "spike-in" controls in a side-by-side comparison of a standard Drop-seq setup and the microfluidic control instrument described here. We analyzed both the sensitivity (molecules/barcode) and accuracy (correlation with known ERCC abundances), and found identical performance between both Drop-seq implementations (Supplementary Figure 3). Taken together, we conclude that our miniaturized device

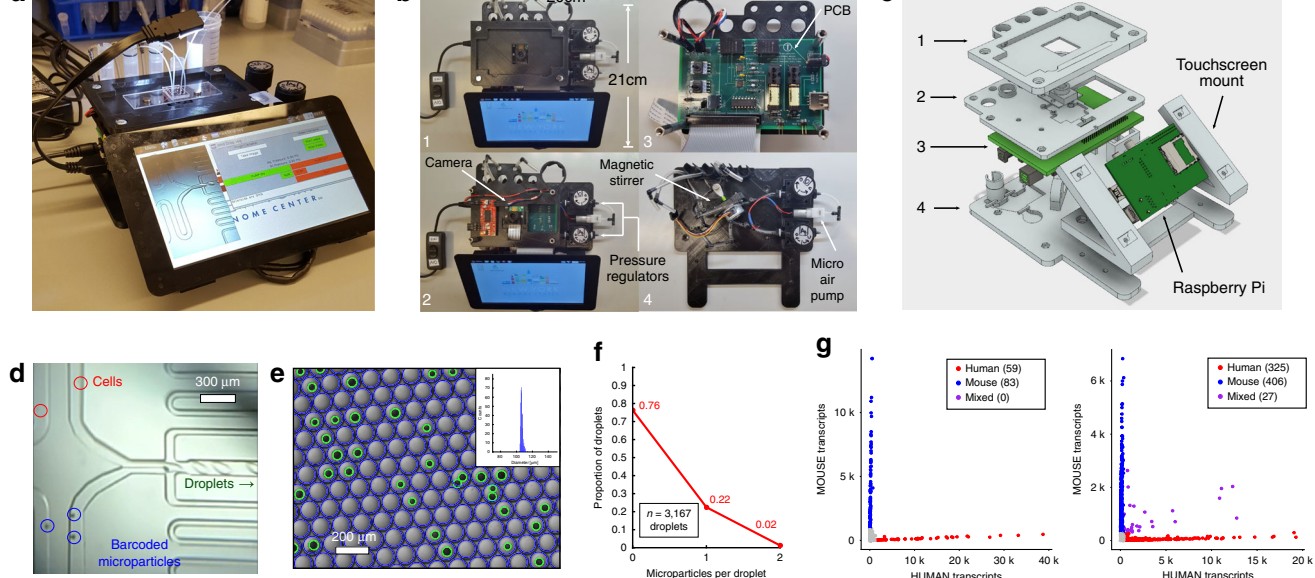

**Fig. 1** Microfluidic control instrument design and validation. **a** Picture of the microfluidic control instrument performing a Drop-seq run. **b** Top down views of multi-tiered instrument. Levels 1–4 contain assorted components for instrument operation. **c** 3D rendering of the instrument with levels corresponding to those in **b**. Components in light gray are 3D printed. **d** Microscope image screen capture directly from the instrument. Cells and barcoded microparticles are visualized easily on the screen. **e** Microscope image of droplets output from the instrument. Droplets and microparticles are detected via image analysis software as blue circles and green circles respectively. Inset: droplet diameter distribution histogram. **f** Microparticle loading distribution into droplets as measured via automated image analysis is consistent with Poisson loading. **g** Species mixing experiments using mouse (3T3) and human (HEK293) cells at total cell concentrations of 75 cells/μl (left) and 300 cells/μl (right)

recapitulates the technical characteristics of a standard Drop-seq setup, but with significantly reduced cost and footprint.

**Transcriptomic profiling of synovial tissue**. After validation of the instrument, we explored the possibility of using our setup to perform Drop-seq directly on patient tissue in a clinical setting. We focused on the inflammatory cellular milieu in synovial tissue extracted from the knees of 5 seropositive RA patients (Fig. 2a). Histologically the RA tissue displayed characteristics of extensive inflammation, including synovial lining hyperplasia (black arrow) and dense leukocyte infiltrations in the sub-lining (blue arrow) (Fig. 2b). Fibroblast morphology within the sub-lining varied widely in the intervening space suggesting that subpopulations of fibroblasts may exist in heterogeneous micro-niches. For each patient, immediately after surgery, a portion of the recovered joint tissue was processed using an optimized disaggregation protocol to generate a single-cell suspension. For two of the patients, we performed side-by-side replicate experiments using identical instruments to assess technical reproducibility. Cells were counted, re-suspended for optimal single cell loading into droplets, and immediately pipetted into the appropriate fluid reservoir of the instrument to run through the Drop-seq protocol. Briefly, following encapsulation in droplets, cells are lysed and mRNAs hybridized to microparticles undergo reverse transcription in bulk to generate stable cell-barcoded cDNAs as previously described[1]. The total time starting with sample extraction from the patient to initiation of the microfluidic instrument is approximately 1.5 h, obviating the need for cell preservation. In total, we collected data from 20,387 single cells, sequenced to an average read depth of 29,651 reads/cell, and detecting an average of 2,315 unique molecules per cell.

We applied our previously described graph-based clustering procedure[10,38], to conservatively partition cells into 13 distinct subpopulations, which we visualized using t-distributed stochastic neighbor embedding (t-SNE) (Fig. 2c, d). While the clustering was unsupervised, differential expression revealed combinations of known markers that could be used to confidently assign subpopulations to broad categories. For example, we observed 10 immune populations that broadly expressed *PTPRC* (CD45) and three fibroblast populations, expressing uniform high levels of *COL1A2*. Similarly, as we explored further within immune cells, we identified clear markers of known subtypes, including canonical macrophage markers (*MARCO*), T cell (*CD3*), and B cell (*MS4A1*) markers (Fig. 3).

We observed that all clusters contained cells from each of the five patients, though we did observe patient heterogeneity in cell type frequency (Fig. 2d, f). However, when comparing replicate experiments from the same patient, we observed tight conservation between the two runs (Fig. 2e; mean $R = 0.98$). Additionally, we compared averaged expression levels for cells in the same cluster across replicates. For example, global macrophage transcriptomes were highly reproducible between replicates ($R = 0.99$) (Fig. 2g), but transcriptomes for different cell populations were widely divergent as expected (macrophage/CD8+ T cell $R = 0.84$ and $R = 0.79$ for RABP3 and RA153, respectively) (Fig. 2h). These results demonstrate the reproducibility of the overall workflow. Additionally, our reproducible and quantitative 'in silico' bulk transcriptomes offer an alternative to traditional bulk RNA-seq on sorted populations, as our procedure requires no sorting, and can derive averages for all 13 populations simultaneously.

To our knowledge, this single-cell dataset represents the first unbiased and comprehensive 'atlas' of cellular subpopulations present in human autoimmune disease tissue. Below, we summarize both abundant and rare cell states in our data, with

unbiased markers shown in Fig. 3 and Supplementary Data 1. We highlight particular subtypes of lymphocyte and myeloid cells that have not been previously identified in healthy PBMCs, as well as unexpected transcriptomic heterogeneity within fibroblast populations.

**Unsupervised taxonomy of cellular states in synovial tissue**. We identified six lymphocyte subpopulations corresponding to heterogeneous groups of T, B, and NK cells. T cells (CD3+) were grouped into CD4+ (helper) and CD8+ (cytotoxic) subpopulations based on canonical markers. Within the CD4+ T helper cell population we detected a distinct subset marked by high levels of *MAF*, *CXCL13*, and *PDCD1* (PD1), which has not been previously identified in previous single cell RNA-seq studies of human PBMCs[4,10]. However, a recent CyTOF analysis of RA synovial tissue identified a population with consistent markers, representing an RA synovial "peripheral T helper cell" ($T_{PH}$) that may support B cell activity and antibody production in this non-lymphoid tissue[39] (Fig. 2c). Pathway enrichment analysis tailored to single cell data[40] identified functional modules up-regulated specifically in these cells, including the regulation of inflammatory cytokine production and B cell differentiation (Supplementary Figure 4), supporting these functional analyses, and demonstrated our ability to identify cellular phenotypes that are relevant to diseased tissue.

Closer inspection of individual groups also revealed further cellular heterogeneity within the T and NK lymphocyte subsets. Within NK cells (uniformly expressing *GNLY*), further sub-clustering revealed a subpopulation expressing high levels of the cytokines XCL1 (lymphotactin) and XCL2, which have previously been demonstrated to regulate fibroblast production of matrix metalloproteinases and direct lymphocyte migration in synovial tissue[41]. This subpopulation also down-regulates cytotoxic genes (*PRF1*) and *FCGR3A* (CD16), representing a bifurcation between CD16+CD56bright and CD16−CD56dim subsets (Supplementary Figure 5). Notably, while CD56bright cells are rare in healthy tissue and have not been identified in scRNA-seq analyses of PBMCs[4], our analyses are consistent with previous reports that the presence of this subset is enriched within RA tissue[42]. Further exploration within all T cells revealed populations that were consistent with the global analysis, but also identified a rare population enriched for the expression of *FOXP3*, *IL2RA* (CD25), and *IKZF2*, likely representing CD4+ CD25+ regulatory T cells, which have also been previously reported to be enriched in inflamed synovial tissue[43].

We also characterized B cell populations (*MS4A1+*) (also known as CD20), as well as terminally differentiated populations that secrete high levels of immunoglobulins (*IGHG4+*). These plasma cells could be further subdivided into two distinct populations based on antibody light chain usage (IgA kappa+ vs. IgA lambda+) (Supplementary Figure 5). This enabled us to calculate a kappa/lambda ratio based on single cell proportions (1.75, 1.98, 3.03) for three patients where we observed at least 25 plasma cells. Finally, we also identified four non-lymphocytic hematopoietic subpopulations, including mast cells (*TPSAB1+*), macrophages (*MARCO+*) and platelets (*VWF+*). Taken together, these clusters represent an detailed and unsupervised characterization of tissue-resident immune cells from inflamed synovial tissue, including both abundant and rare populations that contribute to disease biology.

**Identification and validation of fibroblast subtypes**. Non-hematopoietic cells were primarily composed of subpopulations expressing gene sets consistent with the fibroblast lineage such as *COL1A2*, *COL3A1*, and *CLU* (Figs. 3a and 4a). While immune cell

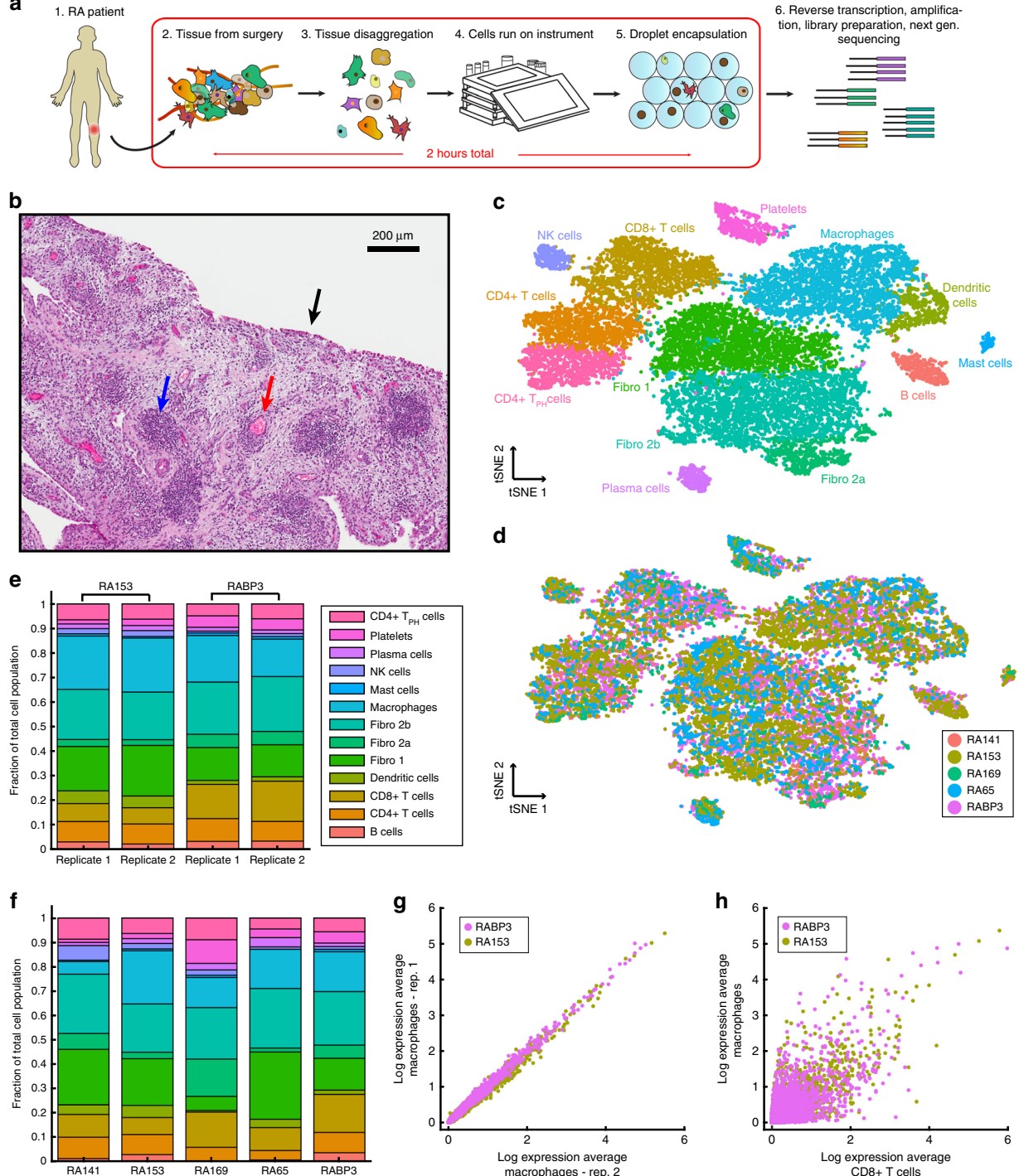

**Fig. 2** RA single-cell unsupervised clustering and analysis. **a** Sample workflow from operating room to sequencing. Preparation of single cells into droplets with barcoded microparticles is performed in about 2 h. **b** H&E stain of synovial tissue from patient tissue. The synovial lining is indicated by the black arrow. An example of vasculature is indicated by a red arrow. The blue arrow denotes a dense lymphocyte infiltrate. **c** Unsupervised graph-based clustering of single-cell RNA-seq, visualized using t-distributed stochastic neighbor embedding (tSNE). Each point represents a single cell (droplet barcode). **d** Graph-based clustering of single-cell RNA-seq, visualized using tSNE colored by patient sample. **e** Fraction of total cells present in each cluster across two patients (RA153 and RABP3) in replicate. **f** Fraction of total cells present in each cluster, for each patient. Color legend is as in **e**. **g** Bulk expression ('in silico average') comparisons across macrophages from each replicate for patients RA153 and RABP3. **h** Expression comparison across combined CD8+ T cell and macrophage populations from both replicates for patients RA153 and RABP3

subsets can be defined based on canonical marker expression, potential sources of cellular heterogeneity in fibroblasts are poorly understood, despite their strong implication in inflammatory disease biology[26–28]. Our unbiased clustering returned three

fibroblast subpopulations (Figs. 3a and 4a, b). These represented two groups of fibroblasts with distinct bifurcations in marker expression (Fibroblast 1 vs. Fibroblast 2), as well as a further subdivision of the latter (Fibroblast 2a vs. Fibroblast 2b)

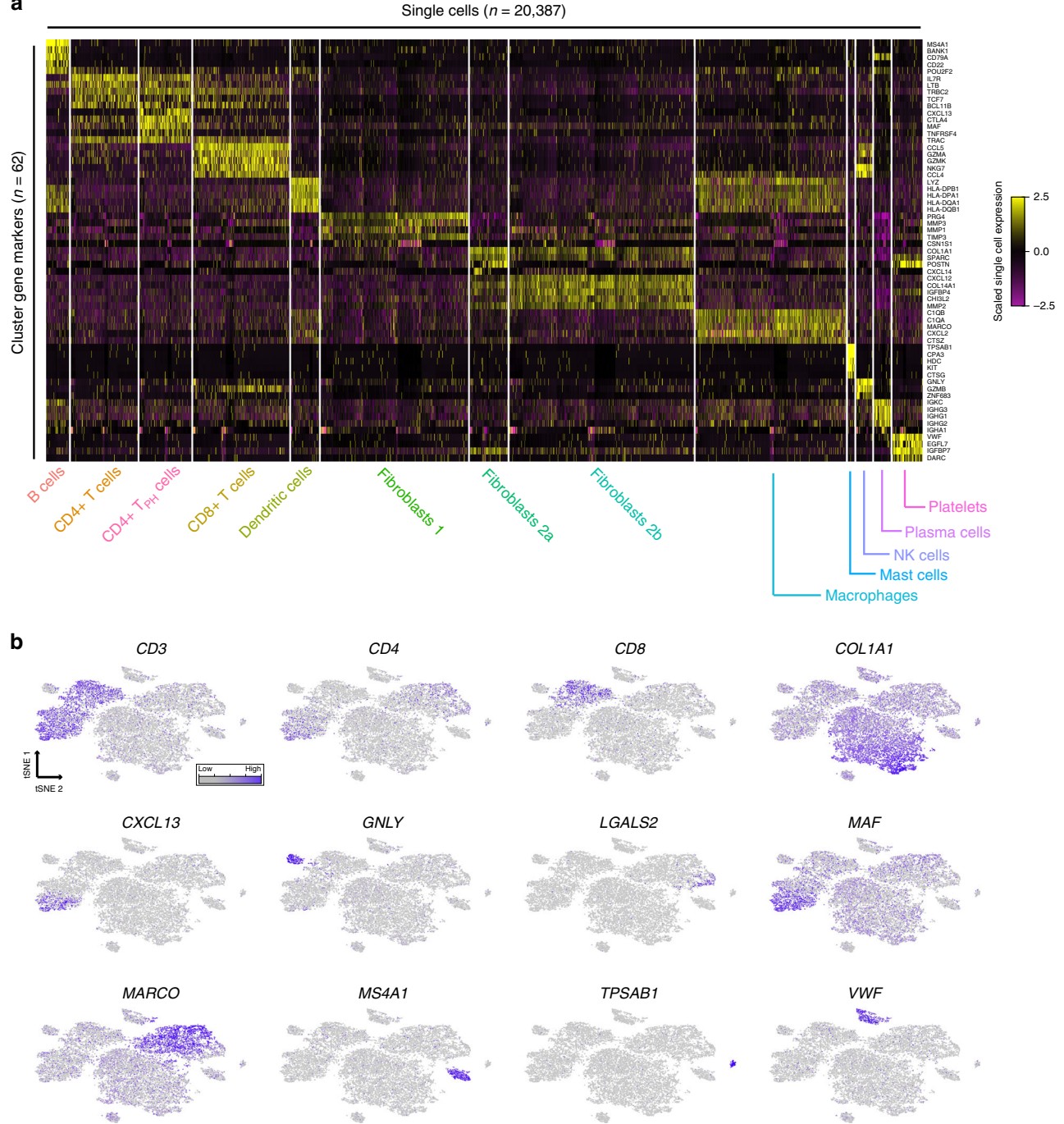

**Fig. 3** Transcriptomic markers of gene expression for individual clusters. **a** Single cell expression heatmap displaying up to five transcriptomic markers for each cluster, based on differential expression testing. **b** Gene expression for canonical marker genes, overlaid on the tSNE visualization. A list of transcriptomic markers for each cluster is provided in Supplementary Data 1

representing more quantitative differences in gene expression. Genes differentially expressed between the subsets included known drivers of RA biology, including cytokines (CXCL12), matrix metalloproteinases (MMP2, MMP3), in addition to a subset of surface protein markers (i.e., CD55; CD90) (Fig. 4a-c).

We next looked to validate the major separation of fibroblast subsets, and to test the specificity of the putative markers using complementary techniques. We first used flow-cytometry of non-hematopoietic viable cells from this tissue to demonstrate that CD90 and CD55 antibodies stained independent cell populations (Fig. 5a, middle panel). The CD55+ cells were largely positive for

the common fibroblast marker podoplanin, while the CD90+ non-hematopoietic cells separated into a podoplanin-positive (fibroblasts) and podoplanin-negative population (Fig. 5a, right panel). The CD90+ CD45− PDPN− population likely represents perivascular stromal cells or endothelial cells[44]. Bulk RNA-sequencing analyses of podoplanin-positive CD90+ vs. CD55+ cells from two patients exhibited highly similar expression patterns when compared to our 'in silico' averaged fibroblast clusters (Figs. 4a and 5b), and genes that were differentially expressed in the population samples showed strong agreement with our original single-cell predictions (Fig. 5c) Therefore, CD90

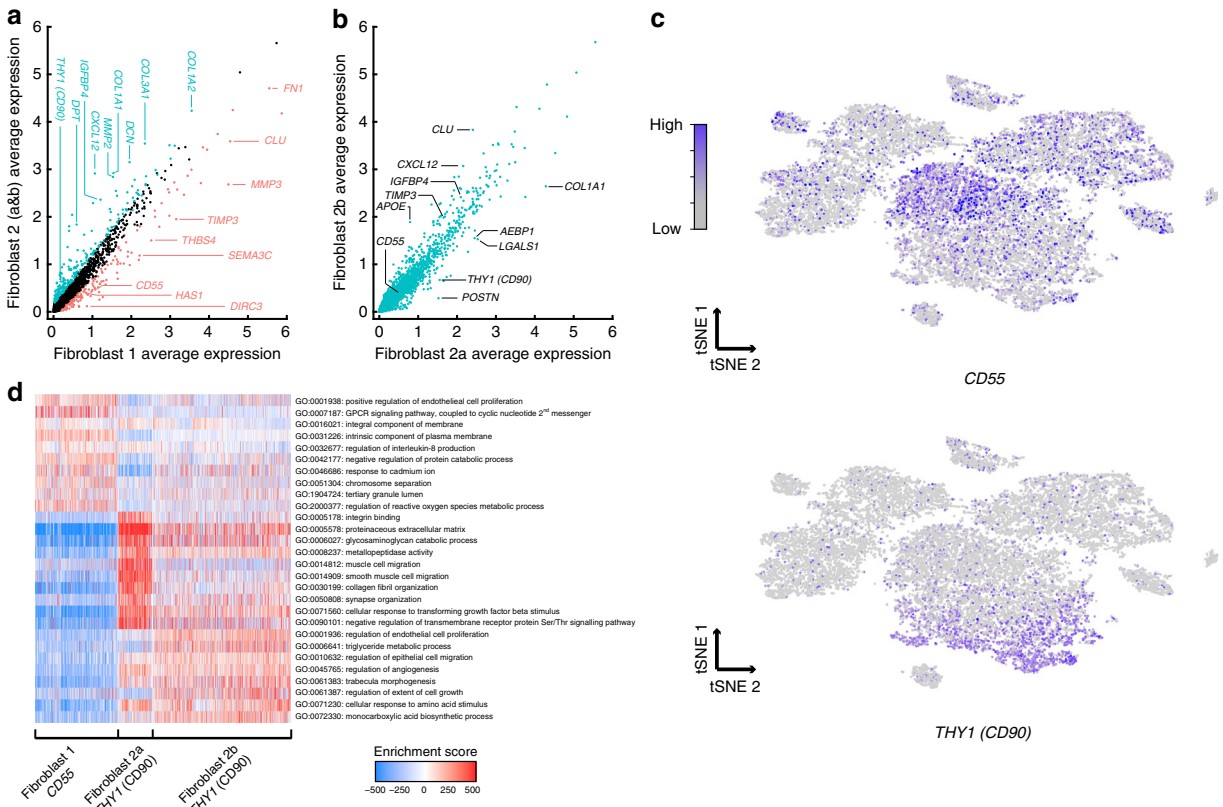

**Fig. 4** Identification of synovial fibroblast subtypes. **a** Bulk expression ('in silico average') comparison of fibroblast populations (1 vs. 2a&b combined across all patients). Genes up-regulated in fibroblast population 1, based on differential expression analysis with Bonferroni-corrected $p < 0.05$, are indicated in red and genes expressed predominantly in fibroblast population 2 are indicated in teal. **b** Expression comparison across fibroblast sub-populations 2a and 2b. **c** CD55 (top) and THY1 (CD90) (bottom) expression across the global tSNE. **d** Pathway and gene set overdispersion analysis on the three fibroblast populations identified from unbiased clustering of single cell RNA-seq data from patient RABP3. Enrichment score corresponds to each cells' first principle component loading from pathway analysis as computed in *pagoda*

and CD55 antibodies specifically mark our Fibroblast 1 and Fibroblast 2 populations, indicating two transcriptomically distinct fibroblast subsets in the RA synovium.

We next asked whether these fibroblast subsets exhibited distinct spatial localization in synovial tissue. To this end, paraffin-embedded tissue blocks for this tissue were sectioned and analyzed by immunofluorescence with antibodies against subset-specific markers (Fig. 5d). Importantly as this approach examines the cells and markers within the intact tissue, it eliminates dissociation-induced artifacts and potentially informs on anatomic localization within the tissue. Interestingly CD55 (Fibroblast 1 marker) predominantly stained in the synovial lining (Fig. 5d). Distinctly, CD90 antibodies (Fibroblast 2 marker) labeled cells in the sublining regions, with intense staining of individual cells disseminated throughout the sublining and intermediate staining that encircled wider rings around larger vessels (Fig. 5d and Supplementary Figure 6).

The distinct anatomical distribution of Fibroblast 1 and 2 populations hereafter referred to as CD55+ lining and CD90+ sublining fibroblasts, respectively, implicate putative functional differences. CD55+ fibroblasts locate to the intimal lining, which is responsible for the generation and turnover of synovial fluid. Importantly, hyaluronan synthase 1 (HAS1) expression was enriched in CD55+ lining fibroblasts (Fig. 4a). As hyaluronan represents the most abundant macromolecule in synovial fluid, this suggests these cells function within the lining to produce synovial fluid components. Pathway and gene-set enrichment analysis revealed hierarchical relationships between fibroblast

subpopulations that were consistent with transcriptomic similarities, and also suggested heterogeneous pathway activation between these groups (Fig. 4d). For example, CD55+ fibroblasts expressed functional modules associated with endothelial cell proliferation and regulation of reactive oxygen species responses, while both CD90+ fibroblast groups were enriched for modules associated with metallopeptidase activity and the organization of the extracellular matrix. Collectively, distinct anatomic locations, cell surface staining and transcriptomic differences confirm the independent nature of these synovial fibroblast subsets.

## Discussion

In this study, we developed a low-cost, portable microfluidic control instrument to perform droplet-based single-cell transcriptomic profiling in a clinical laboratory. Using this instrument we profiled thousands of single cells derived from synovial tissue obtained from 5 RA patients immediately after joint surgery. This methodology allowed us to profile gene expression in a highly quantitative and unsupervised manner across all cell populations simultaneously, providing an attractive alternative to bulk-sorting followed by RNA-seq. Single cell deconvolution of synovial tissue revealed immune subsets including CD4+, B, and NK cells that likely contribute to RA disease etiology through expression of signaling molecules and their interactions with immune and fibroblast populations. Further, single cell transcriptomic signatures identified sources of heterogeneity, specifically in fibroblasts, corresponding to differences in microenvironment and function. Importantly, this dataset can be used to discover and

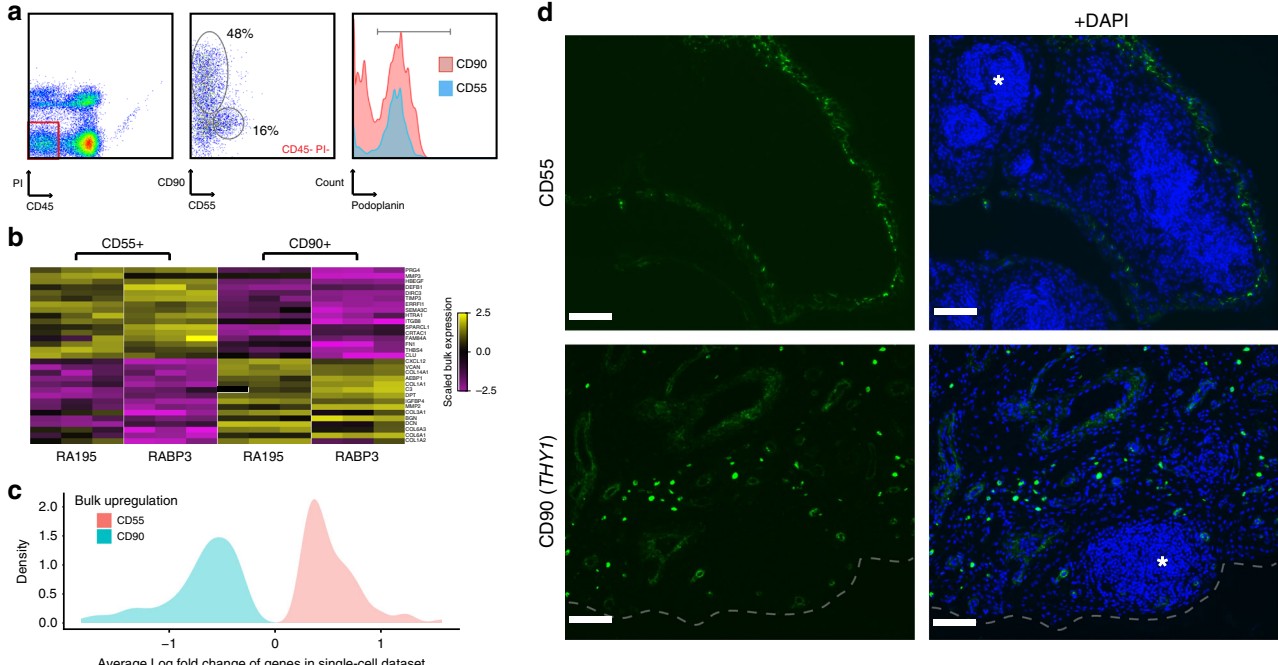

**Fig. 5** Bulk RNA-seq and immunofluorescence of synovial fibroblast subtypes. **a** Cell surface expression of CD90, CD55 and podoplanin in synovial tissue. Synovial cells were gated on the CD45− PI− population (left panel) and analyzed for the proportion of CD90+ and CD55+ cells (middle panel), and relative podoplanin cell surface expression (right panel). **b** Bulk RNA-seq of flow sorted CD55+ and CD90+ patient samples RA195 and RABP3. Heatmap shows that markers of fibroblast populations identified in scRNA-seq data (Fig. 4a) are strongly differentially expressed in bulk. A list of transcriptomic markers for each sorted sample is provided in Supplementary Data 2. **c** We identified genes that were differentially expressed between CD55+ and CD90+ fibroblasts in both the bulk and single-cell datasets. The density of average log fold changes observed for these genes in the single-cell data is colored by upregulation in the bulk data, demonstrating agreement between the bulk and single-cell datasets. **d** CD90 and CD55 localization in RA synovial tissue. Paraffin-embedded synovial tissue was sectioned and assayed for target markers by immunofluorescent staining with antibodies for CD90 (green) or CD55 (green) and counterstained with DAPI (blue). Lymphocyte infiltrates are denoted by gray asterisks. Images were acquired at 20 × magnification. Scale bar is 100 μm

validate putative markers enabling future functional studies. Here we used immunofluorescence to localize CD55+ and CD90+ synovial fibroblasts to the lining and sublining respectively. Information rich datasets obtained from single cell profiling experiments such as this could potentially elucidate disease mechanisms in RA and be used to stratify patients, ascertain treatment efficacy, and classify disease sub-states across many arthritic conditions.

## Methods

**Microfluidic chip fabrication**. Microfluidic chips were designed in AutoCAD (Autodesk) and a transparency mask was manufactured (Advance Reproductions). SU-8 3050 (MicroChem) was spin coated onto a clean silicon wafer to a thickness of 100 μm and exposed through the mask using a contact mask aligner. After development, polydimethylsiloxane (PDMS) was poured over the master mold, degassed in a desiccator and cured in an oven at 80 °C for 2 h. PDMS slabs were cut from the substrate, holes were punched for tubing connections and the slab was bonded to a clean glass slide using oxygen plasma. Finally, microfluidic channels were treated with Aquapel (Pittsburgh Glass Works) and dried in an oven at 80 °C for 30 min.

**Microfluidic control instrument**. The microfluidic control instrument consists of a 3D printed frame affixed with a custom printed circuit board (PCB) designed in Eagle (Autodesk) containing electronic and pneumatic components. The frame accommodates a microfluidic chip viewed at fixed focus through a Raspberry Pi camera with lens. Fluid flow through the microfluidic chip is achieved through pressurization of the head space of reservoir vials situated at the rear of the instrument using a micro air pump, two independent regulators and micro sole-noid valves. Pressures for the oil vial and the aqueous vials (cell and microparticle) were independently measured using two analog gauge pressure sensors. Barcoded microparticles were stirred with a stir disc located inside of the vial under the influence of a permanent magnet affixed to a stepper motor shaft situated at the base of the instrument. The instrument is controlled through a Raspberry Pi 2 model B single-board computer with a custom graphical user interface for

monitoring of the experiment (through the microscope camera) and control of solenoid valves, micro-air pump, and magnetic stirring. The instrument is powered through an external wall adapter power supply (12 V, 5 A) through a barrel jack connection mounted on the PCB. Detailed instructions for the construction and operation of the microfluidic control instrument can be found at https://metafluidics.org/devices/minidrops/[37].

**RA Patient synovial tissue disaggregation**. Synovial tissue was collected from RA patients enrolled and genetically consented under the HSS Early RA Tissue Study (IRB# 2014-317) and the HSS FLARE study (IRB# 2014-233) during syno-vectomy or arthroplasty. The patients were seropositive for CCP antibodies and met 2010 ACR/EULAR Criteria[25]. The HSS Pathology Lab confirmed the sample was synovial tissue by gross inspection and histologic examination of OCT-embedded and Paraffin-embedded blocks.

For single-cell suspensions, synovial tissue was minced with scissors to ~2 mm³ pieces, which were then digested with Liberase TL (100 μg/mL, Roche) and DNAseI (100 μg/mL, Roche) at 37 °C for 15 min with inversion of the sample every 5 min. The enzymatic reaction was quenched by 10% fetal bovine serum in RPMI (Invitrogen) and debris filtered out using two 70 μm strainers. Red blood cells were lysed (reagent a gift of J. Lederer) for 5 min at room temperature, followed by an additional filter step through a 70 μm strainer. The filtration steps should remove large pieces of debris, as well as poorly disaggregated cell clumps. Cells were counted on a hemocytometer and assessed for viability (> 85%) using trypan blue staining and 150,000 synoviocytes were re-suspended in Drop-seq loading buffer.

**Single-cell droplet experiments**. Briefly, cells and split-pool synthesized barcoded microparticles suspended in cell buffer and lysis buffer respectively are co-encapsulated into approximately 1 nL volume droplets. Cell buffer consisted of 1 × PBS with 1.25% Ficoll PM-400 and 0.01% BSA. Lysis buffer consisted of 200 mM Tris-HCl (pH 7.5), 20 mM EDTA, 1.25% Ficoll PM-400, 0.2% Sarkosyl, and 0.01% BSA. Microparticles (ChemGenes) contained oligos consisting of a split-pool generated cell barcode (same for all oligos on a microparticle), a UMI (different for each oligo on a microparticle), a PCR handle and a polyT stretch for capture of polyA mRNA. Following encapsulation (and immediate cell lysis) mRNAs hybridize to the microparticle, the emulsion is broken using 1 H,1 H,2 H,2H-Perfluro-1-octanol (370533 Sigma–Aldrich), microparticles are collected and cDNA is

generated through reverse transcription in bulk. Exonuclease, PCR (15 cycles total), cDNA purification, and Nextera library preparation steps were then performed. Libraries were sequenced on the Illumina HiSeq 2500 platform.

**Single-cell RNA-seq analysis.** Sequencing reads were aligned to the UCSC hg19 transcriptome and then binned and collapsed onto the cell barcodes corresponding to individual microparticles using Drop-seq tools (http://mccarrolllab.com/dropseq). To exclude low quality cells, we filtered out cells for which fewer than 500 genes/4000 UMIs were detected and excluded likely doublets by removing cells with greater than 13,000 UMIs. All genes that were not detected in at least 3 cells were discarded, leaving 30,208 genes. Library-size normalization was performed on the UMI-collapsed gene expression values for each cell barcode by scaling by the total number of transcripts and multiplying by 10,000. The data was then natural-log transformed using log1p before any further downstream analysis with Seurat.

To adjust for the effects of cell cycle, we first assigned a cell cycle score to each cell inspired by the PCA method[45]. We performed PCA on an annotated list of cell cycle genes[46], and observed that the PC1 and PC2 scores for each cell were strongly correlated with the expression of G2/M and S phase modules. We then constructed a linear regression model for each gene to predict expression based on the two cell cycle scores, the number of UMIs per gene, percentage of mitochondrial genes detected, run ID, and alignment rate to the transcriptome. We used the scaled (z-scored) Pearson residuals from this model as corrected gene expression estimates for downstream dimensional reduction. We first selected 2,092 genes with high variance, using the FindVariableGenes function with log-mean expression values between 0.05 and 8 and dispersion (variance/mean) between 0.5 and 30. We then reduced the dimensionality of our data using principle component analysis and identified 13 principle components (PCs) for downstream analysis. We then utilized the Louvain algorithm for modularity-driven clustering[47], based on a cell–cell distance matrix constructed on these PCs. This was implemented using the FindClusters function in Seurat with a resolution of 1 to identify 16 distinct clusters of cells.

We and others[5] have noticed that while modularity-based clustering is a sensitive method for community detection, it can be affected by the multi-resolution problem, and can occasionally over-partition large clusters in order to sensitively detect rare populations. We therefore implemented a post-hoc procedure to merge together clusters with similar gene expression patterns. We reasoned that if a partitioning represented 'over-clustering' of the data, it would be challenging to distinguish the two resulting cluster based on gene expression values. Therefore, for each pair of clusters, we trained a random forest classifier to predict cluster membership based on the expression level of variable genes, using the ranger package in R with default parameters[48]. We merged clusters together if the classifier had a prediction error greater than 11% as measured by the out-of-bag error. This procedure resulted in the iterative merging of two pairs of clusters, both of which also had few differentially expressed genes between them. Finally, we merged two clusters of macrophages that differed primarily in technical metrics of cell quality, including UMI/cell and transcriptomic alignment rate. For visualization, we applied t-SNE on the cell loadings of the previously selected PCs to view the cells in two dimensions.

Our selection of 13 PCs represents a conservative clustering of the data, and we used these to partition our cells into broad subtypes. For immune populations, in particular T cells, NK cells, and B cells, we explored additional sources of heterogeneity by repeating the procedure only on cells from these populations, after the first round of clustering. This enabled us to further separate CD56[bright] from CD56[dim] NK cells, distinguish plasma cells based on IgA kappa+ vs IgA lambda+ expression, and identify rare groups of FoxP3+ regulatory T cells (Supplementary Figure 5).

For all single-cell differential expression tests, we used a non-parametric Wilcoxon rank sum test, as implemented in the Seurat v2.1 package.

**Drop-seq comparison with ERCC spike-in controls.** ERCC RNA spike-in mix (4456740) was purchased from ThermoFisher Scientific. 1 μl of "Mix 1" was diluted in approximately 870 μl of Drop-seq cell buffer and split evenly between side-by-side runs of the standard Drop-seq setup and the microfluidic control instrument described here. Both runs were completed under 15 min. After the runs, droplets were deposited into a Fuchs-Rosenthal hemocytometer and droplet size was measured to ensure identical droplet diameters (volumes). Microparticles were reverse transcribed and exonuclease treated in bulk and counted. One hundred microparticles were pooled for amplification, Nextera library preparation and sequencing in triplicate for each setup. A reference was constructed using the known abundances and sequences provided as supplementary information with the ERCC spike-in kit (https://assets.thermofisher.com/TFS-Assets/LSG/manuals/ERCC92.zip). Reads were mapped to the ERCC reference using the Drop-seq tools pipeline, discarding barcodes with fewer than 100 UMIs. To calculate sensitivity, the total number of detected molecules associated with each barcode was assessed across techniques. To calculate accuracy, the molecule counts for each ERCC were compared to the known molecule abundances in the mix. Pearson correlation (after log-transformation) for each cell independently represents accuracy as defined in Supplementary Figure 3.

**Immunofluorescence.** Antibodies for CD55 (NaM16-4D3) and CD90 (EPR3133) were purchased from Santa Cruz Biotechnology, INC. and Abcam, respectively. Sectioning of paraffin-embedded synovial tissue and immunofluorescent staining was performed by the Molecular Cytology Core Facility at Memorial Sloan-Kettering Cancer Center. The concentrations of CD55 and CD90 antibodies used were 2 μg/mL and 0.7 μg/mL, respectively.

**Flow Cytometry.** Synovial cell suspensions were stained with fluorochrome-conjugated CD45 (H130Biolegend) 1:20 dilution, CD90 (5E10-Biolegend) 1:20 dilution, CD55 (JS11-Miltenyi Biotec) 1:10 dilution, podoplanin (REA446-Miltenyi Biotec) 1:10 dilution, propidium iodide (PI) (P3566 Invitrogen) and analyzed by FACS. Data were analyzed using FlowJo (Tree Star, Inc.) software.

**Flow sorting and bulk RNA-seq.** ~16,000 CD55+ and ~42,600 CD90+ cells were sorted from patient sample RABP3. ~3400 CD55+ and ~11,900 CD90+ cells were sorted from patient sample RA195. Sorting was performed at Weill Cornell Medicine Core Laboratories Center (WCM CLC) Flow Cytometry Core Facility on a BD Aria II sorter. RNA was purified using Qiagen Micro columns and prepared for sequencing using a custom version of the SMART-Seq2 protocol, where we introduce a cell barcode onto the reverse transcription primer, enabling us to pool amplified cDNA from each sample prior to library construction, and described in Mayer et al.[49].

**Pathway enrichment analysis.** We performed pathway and gene-ontology (GO) enrichment for the fibroblast and T cell clusters using the pagoda routines from the scde package on the scaled and normalized scRNA-seq data from each subset separately for the RABP3 patient. We used the genome wide annotation for humans as our reference (Carlson M (2017). *org.Hs.eg.db: Genome wide annotation for Human*. R package version 3.4.1). We performed a PCA analysis and the top principle component for each gene set was obtained using the pagoda.pathway. wPCA function. We then evaluated the statistical significance of each gene set using the pagoda.top.aspects function and retained those with a *p*-value of less than 0.01. To remove redundant GO terms, we used the pagoda.reduce.loading.redundancy function to collapse gene sets driven by the same combinations of genes and the pagoda.reduce.redundancy function to collapse those that separated the same sets of cells. Finally, we took the GO terms with the 10 highest average cell PC1 score for each of our identified clusters for heatmap visualization and analysis.

**Data availability.** RNA sequencing data that support the findings of this study have been deposited in dbGaP with the accession code phs001529.v1.p1.

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

## Acknowledgements

We thank the HSS orthopedic surgeons (particularly Dr. M. Figgie), Dr. Edward DiCarlo of HSS Pathology Department for gross examination and histologic scoring of the synovial sample, rheumatologists, clinical research coordinators (particularly R. Cummings, M. McNamara, and S. Mirza), research technicians (particularly J. Ding, I. Cohn), HSS Research Pathologist Dr. Tania Pellegrini, additional staff in the New York Genome Center Technology/Innovation and Satija Labs, and the consenting RA patients who contributed by providing the tissue used in this study. Research reported in this publication was supported by the following: UH2AR067691 (VB; RBD; LI, with supplemental funding to RS), K01AR066063 (LTD), R35NS097404 (RBD), 5UH2AR067691 (RBD and HPS), R21HG009748 (HPS), NSF Graduate Research Fellowship (DGE1342536 to AB), and an NIH New Innovator Award (DP2HG009623 to RS). RBD is an Investigator of the Howard Hughes Medical Institute.

## Author contributions

W. S. conceived, designed, built, and tested the microfluidic control instrument with input from H. P. S. and R. S. W. S., L. T. D., D. E. O., V. P. B., R. B. D., H. P. S., and R. S. conceived the application to synovial RA tissue. L.T. D. prepared the synovectomy samples and W. S., A. R., and B. B. processed the samples with Drop-seq. W. S. and B. B. performed species mixing and ERCC benchmarking experiments. A. B., W. S., and R. S. analyzed the single cell RNA-seq data. C. R. and L. T. D. performed immunofluorescence and FACS experiments. B. B. performed bulk RNA-seq on sorted cells. S. M. G., L. B. I., V. P. B., D. E. O., R. B. D., H. P. S., and R. S. supervised the research, and provided reagents and funding.

## Additional information

**Competing interests:** The authors declare no competing financial interests.

