## [Peer Review File · Nature Communications]

Reviewers' comments:

Reviewer #1 (Remarks to the Author):

In this study, Stephenson et al. developed a microfluidic control instrument that can be assembled from 3D printed parts. This instrument is highly suitable for a clinical setting due to its minimal costs and its small dimensions. As a proof-of-principle the authors utilize this control device to perform the published Drop-seq protocol on cells isolated from human synovial rheumatoid arthritis tissue. By this strategy they come up with a first comprehensive human cell atlas of an autoimmune disease, comprising immune cells as well as non-immune cell populations.

The device appears to be very useful for enabling a cost-efficient set-up of the Drop-seq protocol as well as other assays utilizing a related technology. Without any doubt large-scale single-cell RNA-seq performed by such a technology would be extremely useful for diagnostic purposes in clinical settings. The control device can facilitate the application of large-scale single cell sequencing in clinical settings as well as basic research labs, which cannot afford expensive commercial solutions. Moreover, to my understanding the setup permits modifications of existing protocols and development of new methods, which is frequently not possible with commercial solutions.

However, I have a few concerns regarding the technology and the analysis that need to be addressed before I can recommend the paper for publication.

1. From the manuscript it is not entirely clear to me, which components are part of the control device. To my understanding, the device comprises the frame, the magnetic stirrer and control electronics. The authors should specify what else is needed to set up the Drop-seq protocol. I'm aware that there might be redundancies with the original publication (Macosko et al., 2015, Cell), but to be really useful it needs to be clear how exactly the protocol can be set up using the control device. I assume that the microfluidic chip and the pumps attached to it for beads and cells are not included in the set up? Are the original specifications from Macosko et al. used in this set-up? What about the microscope that was mentioned in the text? I also did not find templates for the 3D printing included in the manuscript. I think the authors should estimate the total costs including everything required for performing the Drop-seq protocol to enable a comparison with commercial solutions such as 10X Genomics or Dolomite Bio. Finally, the authors should comment on the required expertise for setting-up the device.
2. The validation needs a bit more elaboration. First of all, a doublet rate of 2.57% is reported based on sequencing a mix of mouse and human cells. However, including un-observable doublets of cells from the same species, the actual doublet rate is ~5%. How does the doublet rate depend on the cell concentration and how does this compare to the published protocol in Macosko et al. (2015) Cell? A similar plot to Fig. 3a,b of this paper should be shown.
3. The accuracy of the method should be tested with ERCC spike-in RNA by correlating the measured transcript count to the actual number, and compared to the accuracy of the original Drop-seq method (using the published data).
4. Are there any limitations different from the original Drop-seq method, e. g. regarding cell size? Is the yield (fraction of cells sequenced from input) the same as for the original protocol?
5. Regarding the cell atlas of synovial RA tissue, the authors impressively show that they can recover heterogeneity within immune and non-immune cell sub-populations. While they convincingly show distinct localizations of the two fibroblast populations by immunohistochemistry, they do not investigate co-localization of specific immune-cell subsets with these populations. To further strengthen this part of the story, the authors could, for instance, investigate co-localisation of the CXCL13-positive T helper cell population with each of the fibroblast populations.

Reviewer #2 (Remarks to the Author):

This is a nice written paper on an important problem. However, the key innovation of the paper is related to instrumentation and automation. I am not sure if this is above the bar for Nature Communications. The new knowledge gained in this paper could have been obtained from existing droplet PCR systems. The miniaturization and automation is important and is good engineering but does not add to the science itself. If the authors can justify how this adds to adding new knowledge or allows someone to do experiments they could not before (except for cost and foot print), than I could be more supportive. The paper is more appropriate for an engineering or instrumentation journal.

Reviewer #3 (Remarks to the Author):

In their paper Stephenson and colleagues present the data from single cell RNA sequencing of isolated synovial tissue cells from a single patient with rheumatoid arthritis. There are two separate parts in this paper: the first one is the description of new low-cost platform for single cell RNA sequencing; the second one is the application of this technique in RA synovial tissue.

In general I think that the paper suffers from this mixture of the two aims, Specifically the following concerns are raised:

Part 1: New technology: While the description of the new technique seems appropriate one asks whether it yields similar results as previously used standard Drop-seq setup. For a technology-orientated paper this comparison would seem to be essential in order to judge about the benefit of this novel approach. While the reviewer is not in doubt that this technology is able to differentiate cell populations in the synovium it is difficult to judge the value of this new approach if it is not compared to a current „gold-standard“ approach.

Part 2: Synovial tissue populations: While the data look fine and are interesting, i.e. with respect to the fibroblast subpopulations, one has to consider that the data are derived from one single patient. This is a major limitation of the analysis as we can not be sure whether this single patient is really representative. One would at least want to see such analyses in 5 patients to judge whether the results are representative for RA and are reproducible.

Part 2: Synovial tissue populations: Interesting but one would wish to see a confirmatory approach that shows that indeed two synovial fibroblast populations can be dissected. For instance sorting of CD55 and CD90 fibroblasts could be done followed by subjecting these sorted cells to RNA seq to test whether the defined populations are again identified.

Response to Reviewers' comments:

We thank the three reviewers for their thoughtful and constructive comments. In our revised manuscript, we believe that we have addressed each reviewer concern, and that these revisions have strengthened the work and its conclusions. While we provide a detailed point-by-point response to each comment, below we briefly summarize the new data and analysis included in the revised manuscript, particularly with regards to comments raised by multiple reviewers.

Benchmarking our miniaturized device against a full Drop-seq setup. All reviewers requested a more in-depth comparison of the technical performance of our miniaturized device as compared to the Drop-seq setup first reported in (Macosko *et al.*, Cell, 2015). To address this, we have included two new benchmarking experiments in this revised manuscript. First, we use species-mixing experiments to determine that the 'doublet' rate of our setup is a function of the input cell concentration. These results echo the published findings for Drop-seq. Second, we benchmark the technical quality of the RNA-seq measurements using ERCC 'spike-ins', with known concentration. These results demonstrate that the sensitivity (defined as the number of molecules captured per droplet), and the accuracy (defined as the quantitative agreement between sequenced and known molecular concentrations) are identical between the two setups. Taken together, these new data clearly demonstrate that our miniaturized device can strongly reproduce the technical metrics observed using a full Drop-seq setup, but with a significantly reduced cost and footprint that can enable routine clinical profiling.

Deeper characterization of fibroblast subpopulations in RA samples. Reviewers 1 and 3 requested additional characterization of our single cell RA data set, specifically in regards to the reproducibility of the signals observed across multiple patients and in bulk RNA-seq. To this end we performed additional single cell RNA-seq profiling using the instrument for 4 additional patients. In every patient we observe distinct CD90+ (THY1+) and CD55+ fibroblast populations discerned by our clustering analysis. Next, we performed cell sorting on two patient samples using CD55/CD90 (THY1) markers followed by bulk RNA-seq. Bulk CD55 and CD90 (THY1) cell populations reproduced the expression of the single cell data set indicating that CD55 and CD90 (THY1) are not only transcriptomic markers but also true markers at the protein level that can be used for sorting and enrichment of these fibroblast populations. Finally, we performed immunofluorescence dual staining of CD55 and CD90 (THY1) on an additional RA patient (Also included in the single cell RNA-seq data set) to confirm the lining and sublining localization of CD55 and CD90 (THY1) fibroblast populations respectively.

Description and detailed instructions for users. Reviewer 1 requested more detailed clarification on the requirements for constructing and using the instrument. To clarify the procedure of constructing and using the instrument we have developed a suite of document and materials that fully describe the construction and operation of this device. These documents, alongside a bill of materials, have been included as a supplementary table and can be used for cost comparison purposes between commercial offerings such as 10X Genomics and Dolomite Bio. We envision that with the added documentation and resources the instrument can be constructed with minimal engineering experience and hope that these revisions will enable additional users to take advantage of this low-cost open-source technology.

Reviewer #1

- 1) *“From the manuscript it is not entirely clear to me, which components are part of the control device. To my understanding, the device comprises the frame, the magnetic stirrer and control electronics. The authors should specify what else is needed to set up the Drop-seq protocol. I’m aware that there might be redundancies with the original publication (Macosko et al., 2015, Cell), but to be really useful it needs to be clear how exactly the protocol can be set up using the control device. I assume that the microfluidic chip and the pumps attached to it for beads and cells are not included in the set up? Are the original specifications from Macosko et al. used in this set-up? What about the microscope that was mentioned in the text? I also did not find templates for the 3D printing included in the manuscript. I think the authors should estimate the total costs including everything required for performing the Drop-seq protocol to enable a comparison with commercial solutions such as 10X Genomics or Dolomite Bio. Finally, the authors should comment on the required expertise for setting-up the device.”*

Response: The reviewer raises various concerns regarding the microfluidic control instrument, namely *i)* which portions of the instrument are part of the control device, *ii)* how exactly to set-up the protocol for Drop-seq, *iii)* Availability of 3D printing files for the instrument, *iv)* Cost comparison with commercial solutions and *v)* required expertise for setting-up the device. In order to address the majority of these concerns we have assembled a variety of documents and design files and included them as Supplementary material for this manuscript. Here, readers can find the miniDrops instrument design files (3D printer .stl files, PCB design files, CAD schematic) a bill of materials, user manual, build manual, and python code for completely replicating the instrument and performing a successful Drop-seq run on the miniDrops instrument. The documents, specifically the user manual and build manual address every sub-point *i) – v)* listed above. Additionally, we intend to deposit these documents into a recently established open repository for fluidic systems: metafluidics.org (Kong et al., *Nature Biotechnology*, 2017) to further facilitate dissemination of our work.

- 2) *“The validation needs a bit more elaboration. First of all, a doublet rate of 2.57% is reported based on sequencing a mix of mouse and human cells. However, including un-observable doublets of cells from the same species, the actual doublet rate is ~5%. How does the doublet rate depend on the cell concentration and how does this compare to the published protocol in Macosko et al. (2015) Cell? A similar plot to Fig. 3a,b of this paper should be shown.”*

Response: We thank the reviewer for encouraging us to perform a more extensive benchmarking of doublet rates, as was performed in Macosko et al. We have performed these analyses and included them in the revised manuscript. In particular, we performed multiple human-mouse species mixing experiments at different cell concentrations to accurately quantify the cell loading into droplets. Here we used 75 and 300 cells per μl . We find that our doublet rate is negligible at the lowest concentration, but scales directly with increased cell loading. This in effect reproduces Fig. 3a,b from Macosko et al. Cell, 2015 and we have included this figure as a main figure in the manuscript. Additionally, we encourage users to perform similar species-mixing experiments when initially generating their own data to assess optimal loading concentrations based on their desired doublet rate, as in Macosko et al. Cell, 2015.

- 3) *“The accuracy of the method should be tested with ERCC spike-in RNA by correlating the measured transcript count to the actual number, and compared to the accuracy of the original Drop-seq method (using the published data).”*

Response: Again, we appreciate the reviewer’s suggestion to more thoroughly characterize the technical metrics of our setup in comparison to the full (syringe-pump based Drop-seq setup). As requested, we performed experiments using the ERCC spike-in mix on the standard Drop-seq setup and the miniDrops instrument side-by-side. We assessed both the sensitivity (molecules captured/droplet barcode) and accuracy (correlation of observed sequencing reads and known ERCC concentrations) for both setups in duplicate experiments. We observed indistinguishable accuracy and transcript capture efficiency, and have included these data in the revised manuscript.

- 4) *“Are there any limitations different from the original Drop-seq method, e. g. regarding cell size? Is the yield (fraction of cells sequenced from input) the same as for the original protocol?”*

Response: We envision no differences in the ability of our setup in the proportion or size range of cells captured in our experiments. As we demonstrate in Fig. 1, this is because our device generates droplets with identical size distributions and barcode loading rates as a full Drop-seq setup. Once these droplets have been generated, the downstream processing and molecular biology are fully identical

to Macosko *et al.*, Cell 2015. Taken together, we conclude that our miniaturized device recapitulates the technical characteristics of a standard Drop-seq setup, but with significantly reduced cost and footprint.

- 5) *“Regarding the cell atlas of synovial RA tissue, the authors impressively show that they can recover heterogeneity within immune and non-immune cell sub-populations. While they convincingly show distinct localizations of the two fibroblast populations by immunohistochemistry, they do not investigate co-localization of specific immune-cell subsets with these populations. To further strengthen this part of the story, the authors could, for instance, investigate co-localisation of the CXCL13-positive T helper cell population with each of the fibroblast populations.”*

Response: We thank the reviewer for this comment, and agree that dissecting the close relationship between individual fibroblast populations and specific immune cell subsets is a promising avenue for future explorations. However, we note that these experiments are technically challenging, as they would involve simultaneous profiling of at least 4 protein markers in a single stain (2-3 markers to label and compare specific fibroblast populations, alongside 2-3 markers to label and compare specific T cell subsets). In principle, new technologies based on multiplexed ion beam imaging (Angelo *et al.*, Nature Methods, 2014) would enable to profile this magnitude of markers with spatially resolved resolution, but we felt that these experiments were beyond the scope of the current study. We hope that the reviewer agrees that our efforts to spatially characterize fibroblast populations, alongside their molecular definition, represent a valuable insight for the community.

Reviewer #2

- 1) *“This is a nice written paper on an important problem. However, the key innovation of the paper is related to instrumentation and automation. I am not sure if this is above the bar for Nature Communications. The new knowledge gained in this paper could have been obtained from existing droplet PCR systems. The miniaturization and automation is important and is good engineering but does not add to the science itself. If the authors can justify how this adds to adding new knowledge or allows someone to do experiments they could not before (except for cost and foot print), than I could be more supportive. The paper is more appropriate for an engineering or instrumentation journal.”*

Response: We appreciate the reviewer’s comments, but respectfully disagree with the conclusion that our work will not be significantly enabling for single cell profiling, or that our experiments could have been easily performed with existing systems. We strongly believe that the miniaturization and lower cost entry point represents an advance for the field, and will allow researchers and clinicians to

perform single cell RNA-seq in locations or at throughput previously inaccessible. Indeed, other reviewers recognize and appreciated the value of potential of this work. Additionally, this study represents to our knowledge the first time an autoimmune tissue has been studied at single cell resolution with massively parallel transcriptomic profiling techniques. This represents a model approach and important step forward in translating recent developments in single cell genomic techniques to the clinic to acquire a better understanding of disease. Finally, we see this instrumentation as “open” and modifiable, in contrast to most current single cell offerings. We envision that this instrument may play an important role for the development of novel protocols and techniques in the single cell genomics arena.

Reviewer #3

- 1) *“Part 1: New technology: While the description of the new technique seems appropriate one asks whether it yields similar results as previously used standard Drop-seq setup. For a technology-orientated paper this comparison would seem to be essential in order to judge about the benefit of this novel approach. While the reviewer is not in doubt that this technology is able to differentiate cell populations in the synovium it is difficult to judge the value of this new approach if it is not compared to a current „gold-standard“ approach.”*

Response: The reviewer raises very similar concerns about validation and benchmarking similar to comments raised by reviewer #1 point 2). The experiments outlined in the response to reviewer #1 also address these comments, namely the human-mouse cell loading and ERCC spike-in experiments also address the present comments from reviewer #3.

We thank the reviewer for their comments on our manuscript, and have included a thorough comparison of the technical metrics for our setup in our revised submission. As these concerns were also echoed by the first reviewer, we summarize these revisions at the beginning of this letter. In particular, we show that the doublet rates, sensitivity, and accuracy of our setup and a ‘full’ (syringe pump based) set as reported by Macosko *et al.*, Cell 2015, are indistinguishable. Therefore, we conclude that our miniaturized device recapitulates the technical characteristics of a standard Drop-seq setup, but with significantly reduced cost and footprint.

- 2) *“Part 2: Synovial tissue populations: While the data look fine and are interesting, i.e. with respect to the fibroblast subpopulations, one has to consider that the data are derived from one single patient. This is a major limitation of the analysis as we can not be sure whether this single patient is really representative. One*

would at least want to see such analyses in 5 patients to judge whether the results are representative for RA and are reproducible.”

Response: We thank the reviewer for raising this point, and fully agree that additional patients were necessary to fully substantiate our characterization of new fibroblast subsets. To address this, we have included four additional included rheumatoid arthritis patient single cell RNA-seq datasets to the revised manuscript, increasing our cell total to more than 20,000. Notably, in each of the five patients, we discover cells that fall into either the CD55+ or CD90+ fibroblast subsets, demonstrating that these subsets (and the markers that define them) are both robust and conserved across patients. In addition, cell type frequencies are comparable across patients, but also tightly conserved across replicates. Finally, we performed immunofluorescence dual staining of CD55 and CD90 (THY1) on an additional RA patient (Also included in the single cell RNA-seq data set) to confirm the lining and sublining localization of CD55 and CD90 (THY1) fibroblast populations respectively.

- 3) *“Part 3: Synovial tissue populations: Interesting but one would wish to see a confirmatory approach that shows that indeed two synovial fibroblast populations can be dissected. For instance sorting of CD55 and CD90 fibroblasts could be done followed by subjecting these sorted cells to RNA seq to test whether the defined populations are again identified.”*

Response: While our initial submission reported that CD55 and CD90 marked independent populations of fibroblasts, we agree with the reviewer that we did not provide sufficient evidence that these protein markers could enrich for the exact same populations we identified via single cell RNA-seq. To address this, as suggested, we sorted bulk populations of fibroblasts, after enriching for subsets marked by CD55 or CD90, and performed RNA-seq. We performed this analysis independently for two patients, and compared the differentially expressed genes from our bulk experiment to the markers we observed in our single cell data. As demonstrated in the revised manuscript, the bulk datasets strongly validate our single cell findings, and demonstrate that CD55 and CD90 are valuable markers that can be used to probe the molecular, functional, and spatial characteristics of our newly described Fibroblast subsets.

REVIEWERS' COMMENTS:

Reviewer #1 (Remarks to the Author):

In the revised version of the manuscript, the authors have addressed all of my previous concerns. Most importantly, they provided extensive material for setting up the control instrument and manuals for operating the device. Although I am unable to review every detail of this material it seems to me that it should be possible to set up the system based on these guidelines. Furthermore, they performed the requested control experiments and thereby demonstrated performance comparable to the original Drop-seq pipeline. Since I do think that an easy and cost-efficient set-up of the Drop-seq technology is of interest to a broad community, I recommend the manuscript for publication in Nature Communications.

Reviewer #3 (Remarks to the Author):

I am fine with the answers to my comments. Really think that the authors have done a detailed and in-depth approach to answer them. They conducted new experiments which support their original message. I think the paper is now on much more solid ground.